# The S1P–S1PR Axis in Neurological Disorders—Insights into Current and Future Therapeutic Perspectives

**DOI:** 10.3390/cells9061515

**Published:** 2020-06-22

**Authors:** Alexandra Lucaciu, Robert Brunkhorst, Josef M. Pfeilschifter, Waltraud Pfeilschifter, Julien Subburayalu

**Affiliations:** 1Department of Neurology, University Hospital Frankfurt, Goethe University Frankfurt, 60528 Frankfurt am Main, Germany; waltraud.pfeilschifter@kgu.de; 2Department of Neurology, RWTH Aachen University, 52074 Aachen, Germany; rbrunkhorst@ukaachen.de; 3Institute of General Pharmacology and Toxicology, Pharmazentrum Frankfurt, Goethe University Frankfurt, 60528 Frankfurt am Main, Germany; pfeilschifter@em.uni-frankfurt.de; 4Department of Medicine, Addenbrooke’s Hospital, University of Cambridge, Cambridge CB2 0QQ, UK

**Keywords:** sphingosine 1-phoshate, sphingosine 1-phosphate receptor, S1P_1–5_, sphingosine 1-phosphate metabolism, sphingosine 1-phosphate antagonistst/inhibitors, sphingosine 1-phosphate signaling, stroke, multiple sclerosis, neurodegeneration, fingolimod

## Abstract

Sphingosine 1-phosphate (S1P), derived from membrane sphingolipids, is a pleiotropic bioactive lipid mediator capable of evoking complex immune phenomena. Studies have highlighted its importance regarding intracellular signaling cascades as well as membrane-bound S1P receptor (S1PR) engagement in various clinical conditions. In neurological disorders, the S1P–S1PR axis is acknowledged in neurodegenerative, neuroinflammatory, and cerebrovascular disorders. Modulators of S1P signaling have enabled an immense insight into fundamental pathological pathways, which were pivotal in identifying and improving the treatment of human diseases. However, its intricate molecular signaling pathways initiated upon receptor ligation are still poorly elucidated. In this review, the authors highlight the current evidence for S1P signaling in neurodegenerative and neuroinflammatory disorders as well as stroke and present an array of drugs targeting the S1P signaling pathway, which are being tested in clinical trials. Further insights on how the S1P–S1PR axis orchestrates disease initiation, progression, and recovery may hold a remarkable potential regarding therapeutic options in these neurological disorders.

## 1. Introduction—S1P Metabolism and Signaling

Three decades ago, sphingosine 1-phosphate (S1P) was identified as an intracellular signaling agent in relation to calcium release from intracellular stores and metabolic adaptations [1]. The balance between sphingosine and S1P, both metabolites of its precursor ceramide, and their subsequent activation of effector kinases were shown to matter in imposing regulatory effects in the determination of whether a cell is destined for cell death or proliferation [2]. The sphingolipid metabolism is almost as complex as its protean intricacies to signaling pathways.

### 1.1. De Novo Sphingolipid Synthesis and Signaling at the Endoplasmic Reticulum

De novo sphingolipid biosynthesis is initiated in the smooth endoplasmic reticulum (sER) (Figure 1). Here, the α-aminocarbonic acid serine and the lipid palmitoyl-CoA (PalCoA) are enzymatically processed by the key enzyme serine palmitoyltransferase (SPT)—which is negatively regulated by ORM1-like protein 3 (ORMDL3) [3]—to 3-ketosphinganine [4,5,6]. Subsequent conversion of 3-ketosphinganine to S1P is promoted by enzymatic reactions including a reduction to sphinganine, a synthase reaction to dihydroceramide, and a desaturase reaction to ceramide followed by deacylation by ceramidase (CDase) and a phosphorylation by sphingosine kinase (SphK), which exists in two isoforms (SphK1 and SphK2) [6,7,8]. In general, the formation of the ceramide and the sphingoid bases represents the backbone of the sphingolipid metabolic pathway, as they can be utilized for the synthesis of complex glycosphingolipids. Glycosphingolipids are crucial components of cellular membranes [4,9], such as glucosylceramide or sphingomyelin manufactured by glucosylceramide synthase (GCS) or sphingomyelin synthase (SMS), respectively [10,11]. These various enzymatic reactions are not irreversible per se, since ceramide can be generated by sphingomyelin hydrolysis and/or recycling of complex sphingolipids [10,12,13]. Ultimately, S1P can irreversibly be degraded by S1P lyase into phosphoethanolamine (PE) and hexadecenal, both of which are being further processed [14,15]; PE is used for the synthesis of phosphatidylethanolamine and hexadecenal is used to replenish the PalCoA pool [15,16,17,18]. This cycle of de novo sphingolipid synthesis is tightly controlled by NOGO-B, a protein located within the membrane of the endoplasmic reticulum, which inhibits SPT [19,20]. Alternatively, S1P can be converted back to ceramides by dephosphorylation through sphingosine 1-phosphate phosphatase (SGPP) 1 or SGPP2 [21,22], both of which are members of the lipid phosphate phosphohydrolase (LPP) family [23]. This pathway can substantially contribute to the synthesis of complex sphingolipids within a cell subject to cell type and metabolic demand [24,25].

### 1.2. Synthesis and Signaling of Sphingosine 1-Phosphate in Mitochondria and at the Plasma Membrane

Mitochondrial S1P (Figure 1), produced by SphK2, facilitates oxidative phosphorylation (OXPHOS). This effect was shown to be mediated by stabilizing the scaffolding protein prohibitin-2 (PHB2), which, in turn, eases the recruitment of cytochrome c oxidase (CCO) and thus the assembly of complex IV of the respiratory chain [23,27]. The most important site of S1P production is the plasma membrane (PM) itself (Figure 1), which is composed of a lipid bilayer, predominantly consisting of an extracellular (EC) and an intracellular (IC) phospholipid leaflet. Here, S1P is derived from sphingomyelin, an integral component of cellular membranes [4]. Sphingomyelin is metabolized to S1P via the enzymes sphingomyelinase (SMase), CDase, and sphingosine kinase 1 (SphK1) [28,29]. At this point, S1P can either be exported using the multi-pass membrane proteins spinster homolog 2 (SPNS2) [30,31,32,33] or major facilitator superfamily domain-containing protein 2B (MFSD2B) [30,31,32], respectively, or employed for further immediate intracellular signaling cascades. In terms of intracellular signaling, S1P can engage in tumor necrosis factor-α (TNF-α) receptor-associated factor 2 (TRAF2) [34] and TRAF6-dependent TNF-α signaling [35]. S1P can either recruit the atypical protein kinase C (PKC) subtypes ζ (PKCζ) or ι (PKCι) to sites of distinct membrane microdomains using receptor for activated C kinase (RACK) proteins [36,37,38] or associate with the TRAF2 complex [34]. This, in turn, allows the activation of the interleukin-1 (IL-1) pathway via TRAF6, cell survival, and convergence on the TNF-α pathway downstream of TRAF2 [28,34,39,40,41]. The association with the TRAF2 complex is said to occur by engaging with the N-terminal adjacent really interesting new group (RING) domain, which allows activation of the intramolecular E3 ligase domain of TRAF2 [28,34,41]. The activation of TRAF2′s E3 ligase lures receptor-interacting serine/threonine-protein kinase 1 (RIP1) in close proximity, allowing polyubiquitination of its Lys^63^ residue. Polyubiquitination of Lys^63^ stimulates RIP1′s kinase activity, which results in the phosphorylation of the inhibitor of nuclear factor κ-B kinase (IκB-k). As a consequence, the nuclear factor κ-light-chain-enhancer of activated B cells (NF-κB) signaling pathway is engaged, since IκB-k is now able to facilitate the down-regulation of the NF-κB inhibitor α (IκBα). This culminates in the uncoupling of IκBα from NF-κB revoking its inhibitory effect and allowing NF-κB’s nuclear translocation [28,34,35]. Crosstalk between TRAF2 and TRAF6 was previously reported, as both can engage with atypical PKCs via protein p62 [39], are able to recruit TGF-β-activated kinase 1 and MAP3K7-binding protein 3 (TAB3) [40,42], or can be polyubiquitinylated given the presence of a RING domain in TRAF6 [43]. Moreover, a disruption of TRAF6 binding sites, for example, only mildly impacts NF-κB signaling in the presence of TRAF2 and TRAF3 [44]. Beyond the TNF-α signaling pathway, S1P signaling can vitalize adipogenesis, glucose (Glc) metabolism, and β-oxidation via peroxisome proliferator activated receptor γ (PPARγ) [15,45,46]. By binding to its His^323^ residue, S1P activates PPARγ, increasing the likelihood of an association with the PPARγ co-activator 1β (PGC1β), a necessary co-transcription factor for nuclear translocation [46,47].

### 1.3. Sphingosine 1-Phosphate Signaling in the Nucleus

SphK2, the predominant isoform in the sER and mitochondria [27,48], is also predominant in the nucleus (Figure 1) [49,50,51,52,53]. Nuclear S1P was shown to be critically involved in influencing the balance between cellular quiescence and proliferation. Hait et al., showed that S1P produced in the nucleus binds to the class I histone deacetylases (HDAC) 1 and 2, which results in their inhibition [53]. In general, HDAC1 and HDAC2′s function lies in removing acetyl residues coupled to the α-aminocarbonic acid lysine close to the amino-terminal end of the histone protein H3 [54,55,56]. Therefore, the removal of these negatively charged residues culminates in a net positive charge of H3, increasing its tight association with the negatively charged deoxyribonucleic acid (DNA). Ultimately, the inhibition of HDAC1/2 enhances transcription of cyclin-dependent kinase inhibitor 1 (p21) and the proto-oncogene c-Fos (c-fos) [53]. Conversely, S1P may bind to human telomerase reverse transcriptase (hTERT) [57], which may allow makorin ring finger protein 1 (MKRN1) to dissociate due to competitive binding sites [57,58]. This signaling cascade results in telomere maintenance, cell proliferation, and tumor growth [57].

### 1.4. Sphingosine 1-Phosphate in Autocrine and Paracrine Signaling

Detailed experimental evidence is available for the mechanisms of “inside-out” autocrine and paracrine S1P signaling. After release into the extracellular compartment via SPNS2 [30,31,32,33] or MFSD2B [30,31,32], S1P is swiftly bound by its chaperones due to its hydrophobic character. These chaperones are apolipoprotein M (ApoM)-containing high-density lipoprotein (HDL)—to some extent also to very low-density lipoprotein (VLDL) and low-density lipoprotein (LDL) [59,60]—or albumin, respectively [61,62,63]. Subsequently, upon release and chaperoning by HDL or albumin, ligation of the five known heptameric G protein-coupled S1P receptors (S1PR) 1–5 (S1P_1–5_) (Figure 1) [64,65,66,67,68,69] can result both in autocrine and paracrine signaling [70,71,72]. Signaling via S1PR is tightly regulated. Fine tuning of S1P-S1PR signaling may occur via post-translational modifications, e.g., through palmitoylation by the palmitoyltransferase DHHC5 (DHHC5) [73] and, at some point, termination of the signaling cascade may be achieved by β-arrestin-dependent recruitment of G protein-coupled receptor kinase 2 (GRK2), which phosphorylates S1PR, resulting in dynamin and moesin dependent establishment of the endosome [74,75,76,77,78,79]. At this point, re-routing, i.e., recycling of the receptor to the PM [80], polyubiquitination by the NEDD4-like E3 ubiquitin protein ligase WWP2 (WWP2) resulting in proteasomal degradation [81], or fusion with the lysosome in order for complete proteinaceous and lipid residue degradation can occur (Figure 1) [82,83,84]. The latter endolysosomal salvage pathway is of particular importance for cellular homeostasis and disassembly of complex sphingolipids to ceramides or allowing endolysosomal SphK1 to produce S1P, respectively [85,86,87]. The herein discussed mechanisms are briefly summarized in Figure 1.

### 1.5. External Action of Sphingosine 1-Phosphate through S1PR

With respect to S1PR ligation, the de facto signal triggered is dependent upon the S1PR subtype, the presence of co-regulatory agents, and the heterotrimeric G protein recruited. To date, five bona fide cognate receptors for S1P are known, namely S1P_1–5_ (Figure 2) [88]. S1P_1_, the most commonly expressed S1P receptor in the brain [89], appears to be most selective as it binds only to Gα_i/0_ [66,90]. S1P_2_, also binding to Gα_i/0_, is capable of associating with Gα_q_, G_12/13_, and Gα_s_ [66,88,90], however, it couples most efficiently with G_12/13_, subsequently activating the small GTPase Rho [91,92,93]. S1P_3_ is said to couple with Gα_i/0_, Gα_q_, and G_12/13_, although a higher affinity/likelihood for association with Gα_q_ was observed, ultimately resulting in intracellular Ca^2+^ enrichment and activation of PKC [92,94]. S1P_4_ and S1P_5_ can couple to Gα_s_, Gα_q_, and G_12/13_ [88,95]. Regarding the intracellular signaling pathways triggered, the interested reader is referred to reviews entirely dedicated to detailing molecular signaling and transcriptional cascades triggered in appreciation of the heterotrimeric G protein recruited [96,97,98,99]. Regarding receptor activation or inhibition due to the presence of co-regulatory agents, S1P_1_ can be activated by cluster of differentiation molecule (CD) 44 (CD44) (hyaluronic acid receptor) or activated Protein C (aPC) [100,101], whilst inhibition by CD69, S1P_2_ (in dermal γδ T cells), or LPA_1_ was reported previously [102,103,104]. Unlike NOGO-B, NOGO-A, a multi-pass PM and ER protein whose expression is confined to the central nervous system (CNS), was shown to activate the S1P binding domain Δ20 of S1P_2_, thereby restricting neurite outgrowth via engagement with the G_13_-RhoA signaling pathway [105]. Moreover, conjugated bile acids (CBAs) and FAM19A5 were other activators of S1P_2_ [106,107]. Figure 2 gives a synopsis signaling cascades upon S1P_1–5_ ligation.

## 2. Implications of Sphingolipids in Neurological Disorders

### 2.1. The Sphingolipid Metabolism in Neurodegenerative Disorders

Neurodegenerative diseases are commonly characterized by intracellular or extracellular aggregation of misfolded proteins. These diseases most commonly comprise Alzheimer’s disease (AD), characterized by the proteins amyloid-β and tau, Parkinson’s disease (PD; α-synuclein), and amyotrophic lateral sclerosis (ALS), in which TAR DNA-binding protein 43 deposition is observed [108]. With respect to the accumulation of the ontogeny of the protein misfolded, different classifications of neurodegenerative conditions were established, denoted as tauopathies [109], synucleinopathies [110], or prion diseases [111] (Figure 3).

Regarding AD, increasing evidence supports the crosstalk between sphingolipids and aberrant protein aggregation [112,113]. Amyloid-β-peptide (Aβ) is cleaved from amyloid precursor protein (APP) by the β- and the γ-secretase enzymes, while α-secretase acts within the Aβ sequence [114]. APP cleavage and the release of Aβ from the PM subsequent to its production in lipid rafts are influenced by lipid composition [115,116]. Alterations in membrane lipid composition have a key role in the subsequent subcellular transport and trafficking of these proteins [114,116,117].

Perturbations in the neurovascular unit (NVU) result in a compromised barrier function and dysregulation and reduction in cerebral blood flow (CBF), which is implied to be involved in the pathogenesis of AD [118,119,120,121,122,123,124,125,126,127]. Vascular tightness via tight junctions is influenced by the sphingolipid metabolism. In that regard, acid SMase activity and ceramide production in endothelial cells were linked to vascular permeability [128]. Conversely, the acid SMase inhibition maintained enhanced tight junction regulation [128]. Similar mechanisms were observed to happen in astrocytes [129]. In brain tissue of AD patients’, studies showed increased ceramide levels and decreased sphingomyelin and S1P levels [130,131,132,133,134]. A study on AD by Katsel et al., found a significant up-regulation of messenger ribonucleic acid (mRNA) of phospholipid phosphatase 3 (PLPP3) and S1P lyase 1 (SGPL1) at early stages after diagnosis, suggesting a lack of S1P as a spatiotemporal function may contribute to the degeneration of neurons [135]. Besides, Ceccom et al., reported in an immunohistochemical study of AD a reduction of SphK1 accompanied by enhanced S1P lyase expression in frontal and entorhinal human cortices to be accountable for the perturbed S1P metabolism observed, contributing to the deposition of amyloid and ultimately to neuronal damage [136]. More recently, Dominguez et al., described that the subcellular localization of S1P’s production, e.g., by a disrupted equilibrium between cytosolic and nuclear SphK2, conferred pathogenic effects of S1P in AD [137].

With respect to PD, a chronic progressive disorder characterized by the degeneration of dopaminergic neurons in the pars compacta of the substantia nigra, emerging evidence has elucidated the role of mitochondrial and endolysosomal pathways and their interplay with ceramides in its pathogenesis. Xilouri et al., focused on α-synuclein degradation and suggested, in 2008, an impairment of neural autophagy-lysosomal pathways to be responsible for α-synuclein accumulation, unraveling a causal link between the pathogenic event and the initiation as well as the progression of the disease [138]. Later, further studies confirmed this hypothesis [139,140].

Glucocerebrosidase (GBA) mutations were found in subjects with parkinsonism [141,142], which were previously reported to predispose to the development of Lewy body disorders (LBD) [143]. Concerning LBD, Bras et al., investigated the neuronal ceramide metabolism and reported that several genes known to confer the risk of LBD development converge on the ceramide metabolism, although this remains to be confirmed neurohistologically [144]. In addition, implications in the pathogenesis of synucleinopathies were supported by descriptions of mutations in the GCase gene (*GBA1*) and altered sphingolipid pathways [145,146].

Mazzulli et al., could delineate that glucosylceramidase (GlcCer), the GCase substrates, enhanced the rate of α-synuclein oligomerization [147]. Recently, it was shown that the actual sphingolipid subspecies carry various potentials to cause formation of oligomeric α-synuclein, particularly in reflection of comparing in vitro with in vivo data [148]. Here, glucosylceramide, glucosylsphingosine, sphingosine, and S1P were shown to promote β-sheeted (oligomerized) structures of α-synuclein [148].

Somewhat aside of the molecular establishment of the aforementioned classifications, neurodegeneration is associated with the accumulation of most commonly CAG trinucleotide repeats, which encode for multiple glutamine residues to be translated, inevitably causing a toxic gain of function of the mutant protein [149]. The best-known condition for trinucleotide repeat disorders is Huntington’s disease (HD) [150]. In HD, Di Pardo et al., showed an increase in SGPL1 in the striatum and the cortex and a decrease of SphK1 in the striatum of human post-mortem tissues, which were reflected by similar changes in mouse models of HD [151]. Moreover, the R6/2 mouse model revealed reduced levels of S1P [151,152] despite an up-regulation of SphK2. In contrast, no change was seen in either SphK2 in the YAC128 model or in humans [151]. Unfortunately, no data are currently available regarding S1P levels in YAC128 mice nor in human post-mortem tissues [153]. These findings warrant further investigations into the usability of druggable targets within the sphingolipid metabolism in HD.

Regarding ALS, a study by Henriquez et al., demonstrated a link between ALS severity and gene expressions or metabolite levels for sphingosine, ceramide (d18:1/26:0), SGPP2, SphK1, and UDP galactosyltransferase 8A (UGT8A) [154]. Shedding light on the therapeutic potential of the sphingolipid metabolism in ALS, Potenza et al., reported an improved neurological phenotype and an extended survival after fingolimod, a prodrug that becomes phosphorylated after application in vivo and acts as a receptor agonist against almost unanimously all S1PR—except for S1P_2_ [155,156], administration in mSOD1^G93A^ mice [157].

### 2.2. The Sphingolipid Metabolism in Neuroinflammatory Disorders

Multiple sclerosis (MS) represents an inflammatory disorder of the brain and the spinal cord featuring inflammation, demyelination, and neurodegeneration [158] (Figure 3). Over the last decades, murine experimental autoimmune encephalomyelitis (EAE) models have been used to decipher the mechanisms responsible for disease pathogenesis and progression and to identify druggable targets in order to develop therapeutics for multiple sclerosis [159,160,161]. It has been known for some time that the S1P metabolism can be exploited to slow disease progression in MS, e.g., by fingolimod, causing lymphocyte sequestration and ultimately preventing auto-reactive immune cell infiltration into the CNS [162]. The potency of exploiting the S1P–S1PR axis by fingolimod in EAE was first shown in rats by Brinkmann et al., [155], implicating the feasibility to exploit S1PRs to influence lymphocyte egress. Subsequently, other studies have added to this observation [163,164,165]. Another report unveiled that prophylactic and therapeutic treatment with fingolimod resulted in suppression of EAE [166]. Choi et al., reported, in 2011, that a decline in disease severity of EAE by fingolimod involved astrocytic S1P_1_ modulation as well, thus a loss of S1P_1_ in astrocytes reduced disease severity, demyelination, axonal loss, and astrogliosis [167], arguing for additional CNS-specific effects of fingolimod in addition to lymphocyte redistribution. A recent study identified potential long-term effects caused by S1PR ligation. These long-term effects, according to Eken et al., confer an impact not only on lymphoid sequestration but similarly on non-lymphoid tissue regulatory T cell (T_REG_) distribution, and, more importantly, on reducing the memory T_REG_ pool in favor of effector T_REG_ [168]. This could have implications for appropriate T cell zone access in lymph nodes via C-C chemokine receptor type 7 (CCR7) and subsequently their ability to control auto-reactive T cells in vivo [168,169]. These findings warrant further investigation into the precise mechanisms by which enzymes and lipids involved in the generation of S1P and their effects were linked to disease progression and treatment.

Cruz-Orengo et al., identified S1P_2_ in the inbred SJL mouse strain as a sex- and strain-specific, disease-modifying molecule promoting the breakdown of adherens junctions, thus leading to blood-brain barrier (BBB) leakage, while antagonism of S1P_2_ signaling led to an amelioration of disease severity in female mice [170]. BBB disruption could also be induced by ceramides, resulting in an increased migration of monocytes [129]. Moreover, Lopes et al., demonstrated that acid SMase-derived ceramide regulates intracellular adhesion molecule 1 (ICAM-1) function during T cell transmigration across brain endothelial cells [171].

Concerning S1P receptor expression profiles in the disease model of EAE, mRNA for S1P_1_ and S1P_5_ in the spinal cord was down-regulated, and an up-regulation of S1P_3_ and S1P_4_ mRNAs occurred in the EAE model, which was reversable by fingolimod in accordance with structural restoration of the CNS parenchyma given a restriction to autoimmune T cell infiltration [166]. S1P_3_ was shown to be involved in promoting systemic inflammation via activation of dendritic cells [172]. Concerning S1P_3_ signaling in MS, Fischer et al., suggested that an increased expression of S1P_3_ in EAE was likely due to astrocyte activation; however, its actual sequelae regarding detrimental effects (e.g., astrogliosis) and beneficial effects (e.g., remyelination) could not be established [173]. Apart from astrocytes, the same group reported enhanced SphK1 expression in macrophages of MS lesions [173]. A more definitive proof of S1P_3_‘s importance regarding an inflammatory cascade triggered in astrocytes was reported by Dusaban et al., [174]. The authors demonstrated S1P_3_ to be up-regulated in astrocytes and to be able to engage with transforming protein RhoA (RhoA), and S1P_3_ ligation was shown to promote IL-6, vascular endothelial growth factor A (VEGFa), and cyclooxygenase-2 (COX-2), which was accompanied by an increase of SphK1 and S1P_3_ in vitro [174].

Neuromyelitis optica (NMO) spectrum disorders can also be classified as pertaining to the group of inflammatory brain disorders. Their hallmark feature was initially introduced by Devic and Gault [175,176] and characteristically consisted of a severe complement-mediated damage to the optic nerves and the spinal cord [177]. The discovery of highly specific serum autoantibody marker (NMO-IgG) eventually helped to differentiate this spectrum of disorders from MS and the prior interpretation as one entity [178,179]. Several reports have suggested that treatment with fingolimod in NMO may be contraindicated due to adverse events and worsening of disease severity [180,181,182,183]. However, exploitation of the sphingolipid metabolism to treat patients with NMO should not be excluded prematurely. Matsushita et al., demonstrated significantly higher levels of T_H_1-related, i.e., C-C motif chemokine 4-like (CCL4) and C-X-C motif chemokine (CXC) 10 (CXCL10), and the T_H_-17-related (and neutrophil-related) chemokine CXCL8 (IL-8) in NMO patients [184]. STAT3, which was recently shown to be linked to S1P signaling [185,186,187], is known to control the expression of chemokines and chemokine receptors in the recruitment of neutrophils [188] and T cells [189]. With respect to the mechanisms by which S1P signaling is tied to chemokine production and immune cell migration, studies revealed an interplay between S1PR and chemokine-driven migration of non-activated and naïve T cells [190]. In addition, binding of S1P produced by SphK1 to TRAF2 and cellular inhibitor of apoptosis 2 (cIAP2) in response to IL-1 signaling results in NF-κB activation [34]. This represents a relevant step in the recruitment of mononuclear cells to sites of sterile inflammation by means of interferon regulatory factor 1 (IRF1) expression and the resultant availability of the chemokines CXCL10 and CCL5 [191]. Therefore, mediation of the complex pathways of immune cell recruitment/trafficking, potentially resulting in favorable T_REG_ recruitment without disrupting T_H_-17 or follicular T-helper cell (T_FH_) sequestration, may hold the potential for future S1P metabolism-associated therapeutic perspectives in NMO [192].

Autoimmune conditions, such as systemic lupus erythematosus (SLE) or Hashimoto’s disease, may present themselves with neurological pathology [193,194,195]. Therefore, due to their clinical heterogeneity, affecting potentially any organ of the body such as renal involvement may advise future neurological therapeutic perspectives. In that regard, studies have shown elevated S1P serum levels in patients with juvenile onset SLE [196] as well as in MRL-lpr/lpr mice [197]. In light of therapeutic targets for SLE, previous studies have concentrated on lupus nephritis. Here, fingolimod showed positive effects on survival, suppressing the continuation of autoimmunity [197]. In the context of murine lupus nephritis, in the NZB/W mouse model as well as in BXSB mice, fingolimod also proved to be beneficial [198,199,200]. However, inhibition of SphK2 in the MRL-lpr/lpr model could not convey protection from SLE [196]. These promising results prompted testing of cenerimod, a selective S1P_1_ modulator (NCT02472795).

Other conditions denoted by aberrant inflammatory immune cell activation with a potential to cause immune encephalitis have been reported recently. For example, in autoimmune thyroiditis an enhanced expression of SphK1, S1P, and S1P_1_ converging on STAT3 activation in CD4^+^ T cells was demonstrated in mice by Han et al., [201]. Conversely, administration of fingolimod to these NOD.H-2^h4^ mice conferred the potential to reduce disease severity accompanied by a reduction of STAT3-related cell types, i.e., T_H_1, T_H_17, and T_FH_ cells [201].

### 2.3. The Sphingolipid Metabolism in Cerebrovascular Diseases

Lipid signaling plays pleiotropic roles in cerebral ischemia. In recent years, mounting evidence has emerged depicting the relationship between the sphingolipid metabolism and stroke (Figure 3). Studies have demonstrated that the driving force of neuroinflammation following cerebral ischemia are T cells. They migrate into the brain and amplify the initial detrimental damage [202,203,204,205]. Conversely, lymphocyte-deficient mice were shown to be protected from ischemic damage [206,207].

The S1P analogue and the S1P_1_ functional antagonist fingolimod, originally derived from the fungal natural product ISP-1, was first synthesized in 1992 [208]. It impairs the egress of lymphocytes from primary and secondary lymphoid organs [155] and exerts immunomodulatory effects and non-immunological mechanisms [65,167,209,210,211]. Fingolimod was shown to provide protection from ischemic stroke and intracerebral hemorrhage [89,204,211,212,213,214,215,216,217], leading to the initiation of clinical studies demonstrating the efficacy of fingolimod for patients with acute ischemic stroke and improving clinical outcomes in patients with intracerebral hemorrhage [218]. In addition to reduced infarct volumes and improved neurological scores at 24 and 72 h after middle cerebral artery occlusion (MCAO; a commonly used animal model for ischemic stroke), fingolimod showed a deactivation of caspase-3, a reduction of terminal deoxynucleotidyl transferase-mediated uridine 5′-triphosphate-biotin nick end-labeling (TUNEL-) positive neurons, an activation of RAC-alpha serine/threonine-protein kinase (Akt) and extracellular-regulated kinase (ERK), and a Bcl-2 up-regulation, delineating an anti-apoptotic effect in neurons [211].

Moreover, studies have reported a role of S1PR in the preservation of endothelial barrier integrity [64,219], and phosphorylated fingolimod promotes the establishment of adherens junction in endothelial cells, i.e., an enhanced endothelial barrier function [65,209].

In contrast, Liesz et al., investigated the effect of fingolimod in permanent murine cerebral ischemia without achieving a significant reduction of infarct volumes and behavioral dysfunction despite effective lymphopenia [220]. In addition, Cai et al., unveiled no improvement in functional outcome and BBB integrity in large hemispheric infarctions and administration of fingolimod, either alone or in conjunction with recombinant-tissue plasminogen activator (rt-PA) [221]. Sanchez suggested that S1P_1_ desensitization and/or degradation would potentially evoke detrimental effects on neurons and/or endothelial cells in the context of stroke. Therefore, the dosing and the timing of fingolimod administration seemed to be pivotal for its protective effects [222]. This is in accordance with a previous study by Brait et al., who showed that S1P_1_ fosters protective effects regarding infarct volume after MCAO, however, only if the associated lymphopenia persists for at least 24 h [223].

SphK2 appears to wield an ambiguous nature in various disorders. SphK2 is the predominant S1P-synthesizing isoform in normal brain parenchyma [224] and particularly in cerebral microvascular endothelial cells [225]. SphK2 was recently shown to induce ischemic tolerance to stroke in C57BL/6 mice [226]. SphK2 is preferentially utilized to confer the neuroprotective effects of fingolimod, as it has a 30-fold higher affinity to the prodrug compared to SphK1 [225]. Mice lacking the SphK2 show larger ischemic lesions 24 h after 2 h of MCAO in comparison with wild-type animals [216], thus reinforcing the importance of extracellular signaling of S1PR. Moreover, studies have demonstrated that SphK2 predominates SphK1 in the phosphorylation of fingolimod in vitro [225] and in vivo [227]. In addition, hypoxia increases the expression and the activity levels of the SphK2 isoform in brain microvasculature, subsequently promoting ischemic tolerance [228]. In contrast to SphK2, S1P_2_ is characterized as a key regulator of the pro-inflammatory phenotype of the endothelium [229] and promotes ischemia-induced vascular dysfunction [230]. Conversely, S1P generated by SphK1 potently facilitates the expression of IL-17A in activated microglia, thereby supporting neuronal apoptosis in cerebral ischemia [231]. This is in support of a study by Zheng et al., who found an enhanced expression of SphK1 in microglia 96 h after MCAO [232]. Subsequently, a cortical knockdown of SphK1 resulted in reduced infarct areas and less severe neurological deficits were observed [232].

Studies have highlighted the critical role of S1P_2_ in ischemia-reperfusion injury, confirming that genetic deletion or inhibition of S1P_2_ could block the development of hemorrhagic transformation and cerebral edema by inhibiting the matrix metalloproteinase-9 (MMP-9) activation in endothelial cells [230]. It was shown previously that the use of fingolimod conveyed a reduced risk of hemorrhagic transformation after thromboembolic occlusion [233]. In regard to these findings, the benefit of fingolimod in relation to hemorrhagic transformation was tested in randomized open-label multi-center trials [234,235]. Wan et al., focused on microRNA-149-5p and demonstrated its regulatory function on the permeability of the BBB after transient MCAO in rats by targeting S1P_2_ of pericytes [236]. In their study, the expression of S1P_2_ in pericytes increased at an early stage during ischemia/reperfusion, which was associated with an aggravation of BBB permeability in vivo and in vitro [236]. An engineered S1P chaperone, ApoM-Fc, maintained sustained S1P-S1PR signaling, resulting in a promoted function of the BBB after MCAO [237]. Another study puts emphasis on the importance of S1P in ameliorating the effects of stroke, as they reported reduced S1P lyase activity and a preferential synthesis of S1P and other sphingolipids in response to hypoxia [238].

In opposition to S1P_2_, pathogenic mechanisms of S1P_1_ and S1P_3_ in cerebral ischemia rely on microglial activation [239,240]. Moreover, the same group elucidated the importance of S1P_1_-regulation in promoting a pro-inflammatory M1 polarization of astrocytes, which was brought about by the intracellular signal transducers ERK1/2, p38, and JNK MAPK favoring brain damage after cerebral ischemia [241]. Furthermore, Zamanian et al., examined reactive astrogliosis in response to either MCAO or LPS and showed Pentraxin-related protein PTX3 (PTX3), tumor necrosis factor receptor superfamily member 12A (TNFRSF12A), and S1P_3_ to be markers of reactive astrocytes after MCAO [242]. Liddelow et al., termed them “A1” and “A2” in analogy to the macrophage nomenclature [243]. Interestingly, S1P_3_ was induced 46-fold after MCAO but only 6.4-fold by LPS. Under physiological circumstances, astrocytes were reported to express mainly S1P_1_ and S1P_3_, contrasting with very low levels for S1P_2_ and S1P_5_ [244,245,246]. However, in a recent study by Karunakaran et al., the authors demonstrated the importance of S1P_2_ in microglial activation conferring impaired autophagy and propagating the inflammatory response in the BV2 microglial cell line [247]. Interestingly, similar effects were observed by the group after exogenous S1P administration or genetic knock out of SGPL1 [247].

S1P signaling is also functionally linked to influencing the pathophysiology during subarachnoid hemorrhage (SAH). In accordance with the detrimental effects caused by S1P_2_ ligation in ischemia-reperfusion mentioned before, Yagi et al., demonstrated that S1P signaling increases vascular tone in the context of SAH, thus worsening neurological scores [248]. By employing a selective S1P_2_ antagonistic treatment systemically using JTE013, they were able to set bounds to the extent of myogenic reactivity and to restore neurological scores to sham levels when administered instantly after SAH induction [248].

The various conditions that perturbed S1P signaling has been linked with are summed up in Figure 3.

## 3. Insights into Current and Future Therapeutic Perspectives

Almost a decade ago, the FDA approved the first drug aimed to interfere with the S1P-S1PR signaling cascade, fingolimod, in 2010 [162]. Due to fingolimod’s preference for S1P_1_ and the strong activation of this receptor subtype, S1P_1_ eventually becomes down-regulated, resulting in a long-lasting functional antagonism that accounts for fingolimod-induced lymphocyte trapping in primary and secondary lymphoid organs. Consequently, this lymphocyte sequestration prevents auto-reactive T cells to migrate to the brain and therefore reduces the ferocious neurotoxic damage to myelin-associated proteins in patients with multiple sclerosis [158,249]. Nevertheless, due to its lack of specificity, a range of adverse effects (e.g., first-dose bradycardia [250], macular edema [251], elevated liver enzymes [252], and lymphopenia warranting vigilance regarding occurrence of infections [253,254,255]) is inevitable [256]. The endeavor to circumvent these unwanted drug effects led to the development of more tailored drugs aimed at selectively activating or inhibiting checkpoints within the sphingolipid metabolism on demand. To date, there is a huge number of clinical trials either completed (C), terminated (T), or momentarily being conducted in various clinical conditions examining the pharmacological exploitability of drugs designed to beneficially influence targets within the sphingolipid metabolism. A selection of these studies is presented below (Table 1). It is advisable to conceive that potentially all molecules/agents addressed in Figure 1 and Figure 2 may represent future therapeutic targets.

Safingol, which targets PKC and non-selectively sphingosine kinases (SphK) [257], was identified in 1995 [258]. Safingol is now being tested in various cancer settings, since it is safe to co-administer with cisplatin and exerts tumoricidal effects [259,260]. Similarly, another drug targets SphK. The compound 3-(4-chlorophenyl)-adamantane-1-carboxylic acid (pyridine-4-ylmethyl)amide (ABC294-640) selectively inhibits SphK2. ABC294640 was identified in 2010 [261] and mechanistically competes with sphingosine for binding sites at SphK2. This, in turn, allows sphingosine and ceramides levels to increase (due to inability/slower rate of enzymatical conversion to S1P), facilitating apoptosis-inducing pathways [2,6,262]—a mechanism that was recently studied in various cancer therapies [263,264,265,266]. In that regard, conceivably, the exploitation of PKC and SphK subtypes in neurological conditions where their homeostasis of molecular activation, proliferation, and apoptosis is perturbed by crucially diminishing the S1P level within a cell and its immediate effects via PKC appears intuitive. This remains apprehensible, since S1P levels were previously shown to be enhanced to the disadvantage of its pro-differentiative and pro-apoptotic precursor ceramide [267,268].

Previously, a S1P-directed therapeutic agent was introduced [269]. The S1P-specific monoclonal antibody sonepcizumab (LT1009) is being tested in conditions where pathologies of the vasculature system occur [270,271,272].

Conversely to the aforementioned mechanisms inevitably reducing the amount of S1P available, drugs either mimicking S1P effects at the receptor site or actually increasing S1P levels have been designed. The S1P lyase inhibitor LX3305 is currently being investigated in rheumatoid arthritis as an alternative to therapies with biologicals [273]. Conceptually, LX3305′s tentative application in neurological conditions where S1P is reduced/disturbed appears undoubtedly apprehensible, e.g., in neurodegenerative disorders such as Alzheimer’s disease [134], Parkinson’s disease [274], Huntington’s disease [151,153], or amyotrophic lateral sclerosis [154]. Several diseases have been implicated in aberrant S1PR-specific signaling pathways. In that regard, several drugs specifically designed to interact with S1P_1_ are available to date. AKP11, for example, was compared against fingolimod in a rodent model of multiple sclerosis and was associated with a higher degree of endosomal receptor recycling upon withdrawal, lesser extent of proteasomal degradation, and milder and more easily reversible lymphopenia [275]. Despite similar therapeutic effects, an almost complete absence of adverse events was observed [275]. Similarly, but more recently, BMS-986104 was shown to act equivalently efficient to fingolimod in a T cell transfer colitis model, although not conveying as many cardiovascular and pulmonary adverse events in in vitro settings [276]. Moreover, cenerimod (ACT-334441) could also be confirmed as a potent and selective S1P_1_ agonistic signaling properties, whilst broncho- and vasoconstrictive effects were not clinically relevant [277]. In humans, cenerimod showed an improvement in disease activity scores for systemic lupus without constraining an acceptable safety profile [278]. In contrast to these specific and well-tolerable agents, GSK2018682, another S1P_1_ agonist, did bring about bradycardia and subsequent AV-block [279]. In contrast, BMS-986104 and cenerimod seem to have a favorable risk profile in comparison to fingolimod, which, of course, warrants further investigation in in vivo studies to determine its safety and efficacy in other conditions. Interestingly, in pancreatic islet transplantation, which at least in humans is a definitive treatment for type 1 diabetes mellitus denoted by high mortality and morbidity in the early phase after transplantation [280], KRP203 was shown to cause a marked increase in viable pancreatic islet transplants in C57BL/10 mice [281]. Nevertheless, KRP203 is not an entirely selective S1P_1_ agonist, as it does bind to S1P_3_ with 5-fold and to S1P_2_ and S1P_5_ 100-fold lesser selectivity [281]. Thus, concerns regarding adverse effects, particularly with potentially increased doses necessary and depending on pharmacogenetics, should govern careful investigations in humans. Lastly with respect to S1P_1_ specific compounds, ponesimod (ACT-128800) was previously reported to display therapeutic efficacy in psoriasis whilst also distinguishing itself by swift reversibility upon discontinuation; however, some degree of cardiac effects was detected in clinical trials [282].

It may be advised, under some conditions, to exploit multiple S1PR pathways. S1P_5_, for example, is implicated in having pro-fibrotic effects to act on proliferation and its involvement in early transforming-growth-factor-β (TGF-β)-signaling [283]. Therefore, fibrotic conditions or tissue scarring might be well-suited for treatment with an agent synthesized to evoke agonistic effects both against S1P_1_ and S1P_5_. Three compounds are currently being tested, which act in this pharmacological manner, namely: ceralifimod (ONO-4641), ozanimod (RPC1063), and siponimod (BAF312).

Ceralifimod (ONO-4641), 1-((6-(2-methoxy-4-propylbenzyl)oxy)-1-methyl-3,4-dihydronaphthalen-2-yl)methyl)azetidine-3-carboxylic acid 13n was recently synthesized in 2017 [284]. Beside ceralifimod, ozanimod (RPC1063), which acts similarly, was shown to cause beneficial effects in patients with multiple sclerosis and ulcerative colitis alike [285]. In addition, the authors found that ozanimod wielded strong influence on the innate immune cells. Here, plasmacytoid dendritic cells were lowered (potentially by means of sequestration), which, in turn, reduced interferon alpha (IFN-α) in lupus patients in addition to reducing the entirety of B cell and T cell subsets in the spleen [285].

Lastly, agents that not only depicted S1P_1_ and S1P_5_ agonistic features but that also were partially able to engage with S1P_4_ were identified. This cohort of compounds currently comprises amiselimod (MT-1303) and etrasimod (APD334). The potential of being able to engage with S1P_4_ may hold promising therapeutic benefits. In this regard, S1P_4_, as discussed previously in this review, is predominantly expressed in lymphoid tissues [88,95], and ligation and its subsequent signaling are involved in marking time regarding proliferation [286,287], a reduction of effector cytokines secreted [286,287], and migration of lymphocytes [288,289]. It is worth noting that amiselimod displays a very safe risk profile [290,291,292], while it is too early to consistently assess this for etrasimod [293]. Their application and/or investigation regarding their future therapeutic exploitability in neurological conditions should find due consideration soon.

## 4. Conclusions

In this review, we described the complexities of the sphingosine 1-phosphate (S1P) signaling in neurological conditions in reflection of currently available S1P signaling targeted drugs. Starting with the de novo synthesis of S1P either at the smooth endoplasmic reticulum or other subcellular microdomains, we highlighted the currently established signaling pathways. However, S1P signaling also occurs after secretion and transportation by its chaperones HDL or albumin in the extracellular compartment, allowing either autocrine or paracrine signaling upon ligation to S1P_1–5_. It was highlighted that each S1PR subtype is capable of coupling to a variety of heterotrimeric G proteins, subsequently allowing a tailored intracellular signaling cascade to be incited. However, under perturbed circumstances, the presence of co-activators, inhibitors of S1PR, or simply skewed S1PR patterns may predispose for disease onset and progression. Despite remarkable advances in understanding the contribution of sphingolipid signaling to neurological disorders, the field has yet a lot to learn. In this review, we highlighted the currently available literature regarding perturbations of the sphingolipid metabolism in the context of neurodegenerative, neuroinflammatory, and cerebrovascular diseases. A considerable number of clinical trials are being carried out testing S1P signaling targeted drugs in conditions linked to activation of the immune system. These trials may enhance our understanding of the importance of the S1P–S1PR axis and ultimately help to inform us about future therapeutic usability of these compounds in various neurological disorders. We reported on the importance of S1P_1_ for vascular and other barrier functions. Activation of S1P_1_ causes a significant improvement of vascular barrier properties and prevents microvascular leakage. Currently available drugs interacting with S1P_1_ initially act as agonists but then may cause a profound and long-lasting desensitization and degradation of S1P_1_. As outlined above, they finally act as functional antagonists with, in the long term, negative impact on vascular integrity. Currently, there is no pure S1P_1_ receptor agonist available that does not desensitize the receptor. The compounds described thus far may indeed have a varying degree of agonistic and antagonistic properties. However, such a “true agonist” would be highly desirable and unique in order to protect from vascular leakage.

## Figures and Tables

**Figure 1 cells-09-01515-f001:**
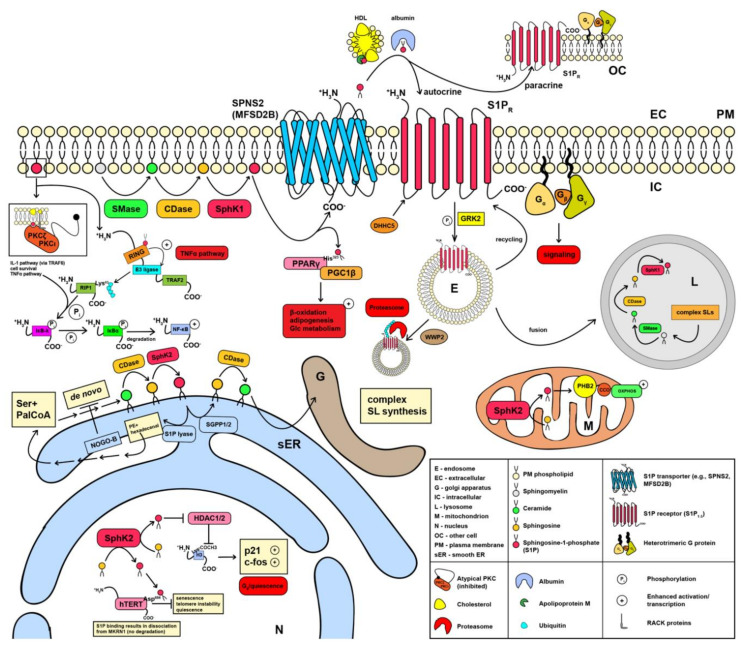
Sphingolipid biosynthesis, sphingosine 1-phosphate (S1P) release, and signaling. S1P is generated in different compartments within a cell. Nuclear S1P influences the balance between chromosome density by histones and telomere length impacting on metabolic adaptations and cell proliferation. De novo S1P synthesized at the smooth endoplasmic reticulum may be utilized for complex sphingolipid synthesis, crucial components of cellular membranes. Mitochondrial S1P influences mitochondrial respiration by activating complex IV. S1P generated at the intracellular leaflet of the plasma membrane (PM) is either used for intracellular signaling converging on the TNFα or the PPARγ pathways or is exported to induce autocrine or paracrine stimulation transported by apolipoprotein (ApoM)-containing high-density lipoprotein (HDL) or albumin and signaling via membrane-bound S1P receptor (S1PR). Intracellularly, S1PR recruit different heterotrimeric G proteins to initiate different signaling pathways, which results in the down-regulation of S1PR via β-arrestin dependent recruitment of G protein receptor β-arrestin-regulated kinase 2 (GRK2), allowing dynamin and moesin-dependent endosome recruitment. Endosomal S1PR are either recruited to the PM or polyubiquitinylated by NEDD4-like E3 ubiquitin ligase WWP2 (WWP2) targeting S1PR for proteasomal degradation. Endosomal remnants are fused with the lysosome (the place where complex sphingolipids are degraded) to fully degrade proteinaceous or lipid cargo, ultimately replenishing the S1P pool. The figure is a modified version of Cartier and Hla [26] and Kunkel et al., [23].

**Figure 2 cells-09-01515-f002:**
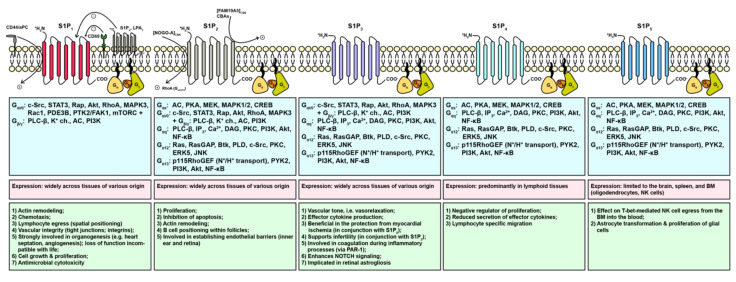
Sphingosine 1-phosphate receptors, canonical pathways, and functions triggered. The currently discovered S1P receptors 1–5 (S1P_1–5_) are displayed. S1PR are naturally activated by S1P and to some extent by dihydro-S1P (sphinganine 1-phosphate) and phyto-S1P (4-hydroxysphinganine 1-phosphate), but also competitive/allosteric activation and inhibition by other molecules are described. Upon activation, S1PR can recruit various heterotrimeric G proteins, which, in turn, allow a finely tuned intracellular signaling cascade to be evoked by means of both Gα and Gβγ. Thus, differential S1PR expression in response to varying environmental lipid contexts, i.e., S1P, dihydro-S1P, and phyto-S1P, respectively, may result in a context- and cell type-dependent function triggered.

**Figure 3 cells-09-01515-f003:**
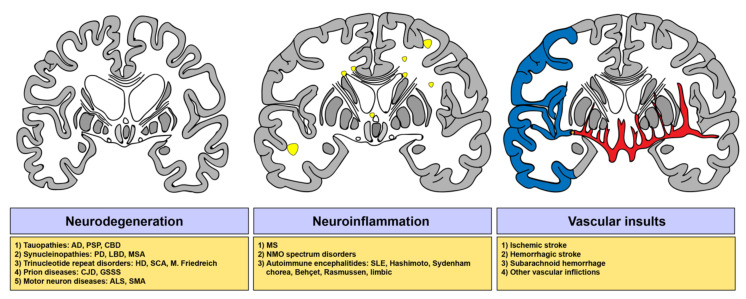
Clinical significance of distorted sphingosine 1-phosphate signaling in neurological disorders. Perturbed S1P signaling has been reported in various clinical conditions ranging from autoimmunity, infection, and cancer. S1P signaling was also shown to play a detrimental role in several neurological diseases, including neurodegenerative and neuroinflammatory conditions, but equally in cerebrovascular insults resulting in stroke or subarachnoid hemorrhage. The yellow patches refer to demyelinated/inflamed areas in the white matter. The area of the grey matter (grey) highlighted in blue denotes the infarcted region. This region is perfused by the middle cerebral artery (here blocked by an embolus), a branch of the cerebral vessel system (red).

**Table 1 cells-09-01515-t001:** Trials of drugs interfering with the sphingosine 1-phosphate metabolism in clinical conditions. (C), completed; (T), terminated; (S), suspended (one study currently on halt due to COVID-19-related recruitment stop); SPHK1/2, sphingosine kinase 1/2; PKC, protein kinase C; S1P, sphingosine 1-phosphate; S1P_n_, S1P receptor ‘n’.

Target	Compounds (Mechanism of Action)	Indications	ClinicalTrials.gov Identifier	Phase
SPHK1/2, PKC inhibitor	Safingol	Solid tumor	NCT01553071	I
		Adult solid tumor (unspec.)	NCT00084812	I (C)
SPHK2	ABC294640 (SPHK2 inhibition)	Non-resectable, perihilar cholangiocarcinoma (extra- and intrahepatic)	NCT03377179	II
			NCT03414489	n/a
		Pancreatic cancer, adult solid tumor (unspec.)	NCT01488513	I (C)
		Multiple myeloma	NCT02757326	I, II (T)
S1P	n/a	Bacterial pneumonia	NCT04007328	II, III
		Food Allergy, anaphylaxis	NCT01776489	n/a
		Asthma	NCT04134351	n/a
		Pneumonia, chronic obstructive pulmonary disease, asthma	NCT03473119	n/a
	Sonepcizumab [LT1009] (S1P-specific mAb)	Solid tumors	NCT00661414	I (C)
		Neovascular age-related macular degeneration	NCT00767949	I
		Exudative age-related macular degeneration	NCT01414153	II (C)
		Pigment epithelial detachment	NCT01334255	I (T)
		Renal cell carcinoma	NCT01762033	II (T)
S1P lyase	LX3305(S1P lyase inhibition)	Rheumatoid arthritis	NCT00847886, NCT01417052	I (C)
			NCT00903383	II (C)
S1P_1_	n/a	Interstitial cystitis	NCT03003845	n/a
		Endometriosis	NCT02973854	n/a
		Vulvodynia	NCT02981433	n/a
	AKP11	Atopic dermatitis	ACTRN12617000763347	II
		Rheumatoid arthritis	ACTRN12617001223325	II
	BMS-986104	Rheumatoid arthritis (healthy volunteers)	NCT02211469	II (C)
	Cenerimod[ACT-334441](S1P_1_ agonist)	Systemic lupus erythematosus	NCT02472795	I, II (C)
		Healthy volunteers	NCT04052360	I (C)
			NCT04255277	I
	CS-077 (S1P_1_ agonist)	Multiple sclerosis	NCT00616733	I (C)
	GSK2018682(S1P_1_ agonist)	Multiple sclerosis (healthy volunteers)	NCT01387217	I (C)
		Relapsing–remitting multiple sclerosis (healthy volunteers)	NCT01466322, NCT01431937	I (C)
	KRP203 (S1P_1_ agonist)	Subacute cutaneous lupus erythematosus	NCT01294774	II (C)
		Hematological malignancies	NCT01830010	I (C)
		Ulcerative colitis	NCT01375179	II (T)
	Ponesimod [ACT-128800] (S1P_1_ agonist)	Multiple sclerosis	NCT02425644	III (C)
			NCT03232073	III
			NCT02907177	III (T)
		Relapsing–remitting multiple sclerosis	NCT01093326	II
			NCT01006265	II (C)
		Plaque psoriasis	NCT00852670, NCT01208090	II (C)
		Chronic graft versus host disease	NCT02461134	II (T)
		Healthy volunteers	NCT02136888, NCT02068235, NCT03882255, NCT02223832	I (C)
		Pharmacokinetics	NCT02126956	I (C)
		Safety and tolerability	NCT02029482	I (C)
S1P_1_, S1P_3_, S1P_4_, S1P_5_	Fingolimod [FTY720] (S1PR modulator, S1P_1_ functional antagonist)	Healthy volunteers	NCT00416845, NCT03757338	I (C)
		Multiple sclerosis	NCT00537082, NCT00670449, NCT00333138	II (C)
			NCT01892722	III
			NCT00662649, NCT00355134, NCT02939079, NCT00340834	III (C)
			NCT01647880	III (T)
			NCT01585298, NCT01333501	IV (C)
			NCT02232061, NCT01981161, NCT02769689	IV
			NCT02139696, NCT01592097, NCT01285479, NCT01281657, NCT02225977, NCT02408380, NCT03216915, NCT01811290, NCT02799199, NCT02776072, NCT01442194, NCT02021162, NCT02307877, NCT03243721	n/a
		Multiple sclerosis (autonomic nervous system dysfunction)	NCT02048072	IV (C)
		Multiple sclerosis (fatigue)	NCT01490840	IV (T)
		Multiple sclerosis (cognitive deficits)	NCT02141022	n/a
		Primary progressive multiple sclerosis	NCT00731692	III (T)
		Relapsing–remitting multiple sclerosis	NCT00289978, NCT01127750, NCT01201356, NCT01497262, NCT01199861	III (C)
			NCT01499667, NCT01633112	III (T)
			NCT01310166, NCT02325440	IV
			NCT02137707, NCT02307838, NCT01420055, NCT02720107, NCT03257358, NCT02373098, NCT01578330, NCT01623596, NCT01534182, NCT01317004, NCT01498887, NCT01216072, NCT01705236	IV (C)
			NCT01755871, NCT02342704, NCT03345940	IV (T)
			NCT01790269, NCT01704183, NCT02335892, NCT02277964	n/a
		Relapsing–remitting multiple sclerosis (cognition, brain volume loss)	NCT02575365	IV (T)
		Relapsing–remitting multiple sclerosis (depression)	NCT01436643	IV (T)
		Acute demyelinating optic neuritis	NCT01757691	II (T)
		Amyotrophic lateral sclerosis	NCT01786174	II (C)
		Intracerebral hemorrhage (hypertensive, intraparenchymal), cerebral edema	NCT04088630	I
		(Acute) stroke, (cerebro-) vascular accident, cerebral stroke	NCT02002390	II (C)
		Chemotherapy-induced peripheral neuropathy (numbness, pain, tingling)	NCT03943498	I
		Chronic inflammatory demyelinating polyradiculoneuropathy	NCT01625182	III (C)
		Breast carcinoma	NCT03941743	I
		Glioblastoma, anaplastic astrocytoma	NCT02490930	I (C)
		Coronavirus disease (COVID-19)	NCT04280588	II
		Asthma	NCT00785083	II (C)
		Rett syndrome	NCT02061137	II (C)
		Schizophrenia	NCT01779700	II (C)
		Renal insufficiency	NCT00731523	I (C)
		Kidney transplantation	NCT00239902, NCT00239798	II (C)
			NCT00099736, NCT00239876, NCT00239811, NCT00099801, NCT00099749, NCT00239863, NCT00239785, NCT00098735	III (C)
S1P_1_, S1P_5_	Ceralifimod [ONO-4641] (S1P_1,5_ agonist)	Multiple sclerosis	NCT01081782	II (C)
			NCT01226745	II (T)
	Ozanimod [RPC1063] (S1P_1,5_ agonist)	Multiple sclerosis	NCT02797015	I (C)
			NCT02576717, NCT04140305	III
			NCT02294058	III (C)
		Relapsing–remitting multiple sclerosis	NCT01628393, NCT02047734	III (C)
		Ulcerative colitis	NCT01647516	II
			NCT02435992, NCT02531126, NCT03915769	III
		Crohn’s disease	NCT02531113	II (C)
			NCT03467958, NCT03464097, NCT03440372, NCT03440385	III
		Healthy volunteers	NCT02994381, NCT03694119, NCT03644576, NCT03624959, NCT03665610	I (C)
			NCT04149678, NCT04211558	I
	Siponimod [BAF312] (S1P_1,5_ modulator)	Healthy volunteers	NCT00422175	I (C)
		Multiple sclerosis	NCT03623243	III
		Relapsing-remitting multiple sclerosis	NCT01185821, NCT00879658	II (C)
		Secondary progressive multiple sclerosis	NCT01665144	III
			NCT02330965	n/a
		Polymyositis (, dermato-myositis)	NCT01801917, NCT01148810, NCT02029274	II (T)
		Hepatic impairment	NCT01565902	I (C)
		Hemorrhagic stroke, intracerebral hemorrhage (ICH)	NCT03338998	II (S)
		Renal impairment	NCT01904214	I (C)
S1P_1_, S1P_5_, (S1P_4_)	Amiselimod [MT-1303] (S1PR modulator, S1P_1_ functional antagonist)	Relapsing–remitting multiple sclerosis	NCT02193217, NCT02310048, NCT02293967	I (C)
			NCT01890655, NCT01742052	II (C)
		Crohn’s disease	NCT02148185	I (C)
			NCT02389790, NCT02378688	II (C)
		Systemic lupus erythematosus	NCT02307643	I (C)
		Plaque psoriasis	NCT01987843	II (C)
		Inflammatory bowel disease	NCT01666327	I (C)
	Etrasimod [APD334]	Primary biliary cholangitis	NCT03155932	II (T)
		Inflammatory bowel disease(extra-int. skin manifestations)	NCT03139032	II (T)
		Ulcerative colitis	NCT02447302, NCT02536404	II (C)
			NCT03950232, NCT03945188, NCT03996369, NCT04176588	III
		Pyoderma gangrenosum	NCT03072953	II (T)
		Crohn’s disease	NCT04173273	II
		Atopic dermatitis	NCT04162769	II

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
