# Peer review of "The S1P–S1PR Axis in Neurological Disorders—Insights into Current and Future Therapeutic Perspectives"

_cells, 2020, doi:10.3390/cells9061515_

Round 1

Reviewer 1 Report

This review discusses the important implication of sphingosine-1-phosphate (S1P) signaling in neurodegenerative and neuroinflammatory diseases. It highlights the crucial role of this lipid via its secretion in the extracellular compartment and its transport through HDL or albumin, allowing autocrine and / or paracrine actions after ligation to its receptors on cell surface. Perturbation of these receptor activities can lead to the development and progression of neurological diseases and the use of agonist/antagonist drugs could improve the treatment of these diseases.

The review summarizes well at fist the complex metabolism leading to the production of S1P in cells and to its no less complex mode of action through the activation of its 5 receptors, allowing a plethoric action of the lipid in an autocrine and paracrine fashion.

I have to admit that the first chapter “S1P metabolism and signalling” is very interesting but it could be enhanced by cutting it into different sub-chapters. This would make it much clearer and easier to read.

The second chapter “Implications of sphingolipids in neurological disorders” is very interesting and sums up our knowledge of the subject. It updates the many discoveries made in recent years.

Finally, the last chapter talks about everything we know about drugs regulating the action of S1P in various neurological diseases.

Overall, it is a very interesting review and can serve as a bibliographic basis for all specialists in the field, but also for students.

Table 1 is interesting but could perhaps be simplified.

Reviewer 2 Report

"The S1P-S1PR axis in neurological disorders - Insights into current and future therapeutic perspectives” by Lucaciu et al.

The topic of the manuscript addresses an interesting issue: the role of S1P-S1PR axis in neurological disorders. This is a really interesting topic since the discovery of S1P receptor and their potent involvement in various diseases with the development of various agonist/antagonist as regulator of their cellular fonctions. The review is well constructed with 4 chapters. However the authors should improve the reading inside some chapter by separating different section to be easy to read and give more information. See below my comments:

1- Introduction:

Line 55, the authors should explain the nature and function of NOGO-B

Line 59, ref 24 to be corrected

line 60 “Mitochondrial S1P…” seems to describe function of S1P whereas the beginning of this paragraph is dedicated to the biosynthesis of sphingolipid with a final focus on S1P metabolism. Then the authors came back to the regulation of secretion of S1P, focused on intracellular effect of S1P to finally write about effect of secreted S1P.

It looks like these chapter is difficult to read and should be well separated in sub-paragraph, where the authors will describe 1/ sphingolipid/S1P metabolism, 2/intracellular effect of S1P and 3/ secretion of S1P and external action through S1PR. This will help reader to follow the introduction and understand the multiplicity of S1P’s action.

Line 119: the figure 1 should be cited a long the introduction and not put at the end of the introduction. Same comment for the figure 2.

2- Implication of sphingolipid in neurological disorders

Line 192: the sentence should be revised by the authors

Line 195: in addition to the regulation of SphK1 in Alzheimer disease, the regulation of SphK2 has been also associated to these disease. See the reference:

Neuronal sphingosine kinase 2 subcellular localization is altered in Alzheimer's disease brain.

Dominguez G, Maddelein ML, Pucelle M, Nicaise Y, Maurage CA, Duyckaerts C, Cuvillier O, Delisle MB. Acta Neuropathol Commun. 2018 Apr 3;6(1):25. doi: 10.1186/s40478-018-0527-z.

Line 210 : the authors describe a role of sphingolipid in Parkinson diseases but with no mention of S1P. If this is the case, the authors should specify it.

Line 215: the last sentence should be clarify. What the authors mean by “alterations”.

At the beginning of this chapter, the authors talk about ALS but later on, there’s no mention of a potent role of sphingolipid or S1P in this neurodegenerative disease.

3- Neuroinflammation

Line 234: “A recent study identified potential long-term effects caused by S1PR ligation. What the authors mean by “long-term effects”?

Line 270 to Line 282: in these line, the authors developed a long hypothesis about the potent role of S1P on inflammation that could be related to the NMO disorder. This should be reduced since this is only speculative.

Line 292: the authors should clarify the last sentence. What do they mean or expected from this assay?

4-Cerebrovascular insults

This chapter is dedicated to the potent therapeutic role of S1P with no explanation of the role of S1P in these disease.

Line 304: it is surprising that the authors mention the origin of fingolimod here since it was already used before (without description of its nature) in the review (chapter 2.2). This should be corrected.

Line 307: “That said” is not necessary

Line 347: The authors should clarify the sentence ‘”while Zamanian et al. examined…”

Line 352: The figure 3 should be cited a long the paragraph. Moreover, the authors should describe the meaning of the various colors used.

5- Insights into current and future therapeutic perspectives

Line 361 to line 364: here, the authors describe the function of fingolimod. This appears to be late since the authors already used data on fingolimod without describing its origin and pharmacological’s action.

Line 371: reference 229/231 should be corrected

In this chapter, the authors describe the potent role of agonist/antagonist/inhibitor in neuronal disorders. However, it is really difficult to follow this description without knowing the pharmacological function of these compounds. The authors should clearly define (not only in the table), the properties of the compound used before describing its potential therapeutic effect.

Reviewer 3 Report

Lucaciu and colleagues set out to summarise the current knowledge in the field of S1P signaling related to neurological disorders. 

Overall, the proportion of the article describing the S1P metabolism (5 of 15 pages) is considerably high considering the title and purpose of the review. The authors explain S1P's metabolism and receptor signaling in great detail but miss to elucidate upon pathological mechanisms with the same level of detail. 

The subheading “Neurodegeneration” is very misleading as neuronal degenerative actions of S1P or its signaling components are not described in the respective text. Instead, the authors list S1P-related findings in different neurodegenerative diseases. Most of them are not even described and still referred to as mechanisms (e.g., 187: Similar mechanisms were observed to happen in astrocytes [123].).

The authors should consider revising their section titles. It is unclear why a section that follows “neuroinflammation” is called neurovascular insults and starts with a statement about the inflammatory responses in stroke and continues in that line of thoughts.

It is unclear what the author mean by: “Examining the effects of sphingolipids on the blood-brain barrier (BBB) function, studies could demonstrate an increased acid SMase activity and ceramide production in endothelial cells after having been exposed to an inflammatory stimulus.” (line 184)

It is unclear how the statements in lines 187-190 link to neurodegenerations, considering the next paragraph called “neuroinflammation”: Blood-brain barrier disruption induced by ceramides led to an increased migration of monocytes [123]. Moreover, Lopes et al. demonstrated that acid sphingomyelinase-derived ceramide regulates intracellular adhesion molecule 1 (ICAM-1) function during T cell transmigration across brain endothelial cells [124].”

Overall, it is misguiding that the “neurodegeneration” section mainly consists of basic descriptions of neurodegenerative diseases and only two paragraphs (max 50%) evolves around S1P signaling in neurodegenerative diseases.

Considering the scarcity of in vivo S1P signaling in stroke, the authors should include all studies in the topic. Important findings related to fingolimod and hemorrhagic transformation as well as S1P signaling component expression after MCAo are missing in the stroke paragraph. Moreover, findings regarding the role of mural S1PR2 in the cerebrovasculature in SAH are not mentioned.

It is unclear what the authors mean by the following: “One of the hallmarks of SphK2-dependent S1P production is neuroprotection. SphK2 is required for the protective effect of phosphorylated fingolimod in ischemic stroke. Mice lacking the SphK2 show larger ischemic lesions 24 hours after 2 hours of MCAO in comparison with wild-type animals [205], thus reinforcing the importance of extracellular signaling of S1PR. (lines 329-332)

Overall, the conclusion does not match the title (purpose?) of the review and reflects the not very well developed storyline of the manuscript.

Round 2

Reviewer 2 Report

"The S1P-S1PR axis in neurological disorders - Insights into current and future therapeutic perspectives” by Lucaciu et al.

The topic of the manuscript addresses an interesting issue: the role of S1P-S1PR axis in neurological disorders. This is a really interesting topic since the discovery of S1P receptor and their potent involvement in neurodegenerative diseases with the development of various agonist/antagonist as regulator of their cellular functions. The authors responded to almost my comments and the revised version is much more easily to read/understand. However there’s still some inaccuracy and missing information to be complete. Finally, the last chapter “Insights into current and future therapeutic perspectives” is interesting but should be more connected to the treatment of neurodegenerative diseases. See below my comments:

1- Introduction:

Line 62: The authors described the de novo synthesis of ceramide which take place in ER but finished by the action of ceramidase and sphingosine kinase to produce S1P. Even this is true, this sentence is mislocalized in the description of the sphingolipid metabolism. The authors should focus first on the metabolism of ceramide into SM and glycosphingolipid. The sentence on S1P should be linked to the line 71 (Ultimately,…) where the authors started again to talk about S1P.

Line 67: I guess that the authors want to talk about glucosylceramide synthase (GCS) and not glycosylceramide synthase. In fact, the synthesis of glycosphingolipid is more complex than only the existence of GCS since cells could produce gangliosides and sulfatides. Maybe, to be complete the authors could modify these part of the introduction.

There’s a description of S1P metabolism in mitochondria, PM and nucleus but not in the ER. This is discrepancy with the description made on the chapter “de novo sphingolipid…”.

Line 112-113 and 127-129: in these lines, the authors mentioned the role of PPARg in S1P signalling. For more clarity, it will be better to link these two sentences and separate them from the description of the role of TRAF in S1P signalling.

Line 116: In one sentence, the authors wrote “This, in turn, allows the activation of the interleukin-1 (IL-1) pathway via TRAF6, cell survival, and convergence on the TNF-alpha pathway downstream of TRAF2.” Could the authors be more specific on the potential cross-talk between in TRAF2 and TRAF6 from this sentence?

 Line 122-127: there’s a very long sentence : 5 lines. Should be cut for more clarity.

Line 268: there’s a repetition “hydrophobic character” in the sentence. Should be deleted.

Line 271-272: the authors mentioned that secreted S1P could be take in charge by apoM/HDL and albumin and specify that both complex could be activators of S1PR. However, albumin/S1P seemed to be a reservoir of S1P but not a signalling molecule. Could the authors specific this notion.

Line 281: Reference 81: is not the correct article cited in the text.

Line 463: Could the authors specify the role of NOGO-A compared to NOGO-B. Do they have the same subcellular localization? If yes, how NOGO-A could regulated S1P1 signalling?

How S1P2 could inhibit S1P1? From the publication cited by the authors, that seems not the case.

Chapter 2

Line 525-526: could the authors clarify this sentence: “Inspired by the demonstrated glucocerebrosidase (GBA) mutations in subjects with parkinsonism [135,136], Bras et al. addressed neuronal ceramide metabolism in Lewy body diseases [137].

Line 764: the authors should define MCAO in the text.

Chapter 3:

This chapter is interesting but reading it, it’s looks like somewhat disconnected from the rest of the review. The authors described the effect of new compounds (especially S1PR agonist and antagonist) used in different clinical trials not link to neurodegenerative diseases. Since the authors nicely describe the role of Fingolimod (and its limits) on neurodegenerative diseases, it will be important to make some comment on the usefulness of these new compound for the treatment of the neurodegenerative diseases listed in the Chapter 2.

Reviewer 3 Report

The authors have satisfactorily responded to all comments.

For point 5 raised in the initial review, the authors requested the references for the SAH related publication: https://doi.org/10.1161/STROKEAHA.114.006365
